# Analytical Considerations of Large-Scale Aptamer-Based Datasets for Translational Applications

**DOI:** 10.3390/cancers14092227

**Published:** 2022-04-29

**Authors:** Will Jiang, Jennifer C. Jones, Uma Shankavaram, Mary Sproull, Kevin Camphausen, Andra V. Krauze

**Affiliations:** 1Radiation Oncology Branch, Center for Cancer Research, National Cancer Institute, NIH, 9000 Rockville Pike, Building 10, CRC, Bethesda, MD 20892, USA; will.jiang@nih.gov (W.J.); uma@mail.nih.gov (U.S.); sproullm@mail.nih.gov (M.S.); camphauk@mail.nih.gov (K.C.); 2Translational Nanobiology Section, Laboratory of Pathology, NIH/NCI/CCR, Bethesda, MD 20892, USA; jennifer.jones2@nih.gov

**Keywords:** aptamers, biomarkers, proteomics, bioinformatics, translational

## Abstract

**Simple Summary:**

Aptamers represent an emerging technology that enables researchers to screen biological matrices such as blood and urine for thousands of different proteins at a rapid pace with high precision and accuracy. However, the sheer data volume generated by this high-capacity screening technique also creates a fundamental challenge towards efficiently analyzing these complex datasets and translating findings for the clinic. We address the new analytical considerations brought forth by aptamers, explore the necessary statistical analysis needed, and create a baseline to analyze these large-scale databases more comprehensively. In addition, we explore how aptamers can co-exist with current proteomic platforms to produce more robust findings in an evolving, multi-faceted approach towards the field. Unlocking the underlying signals masquerading behind these large datasets will ultimately empower clinicians and researchers to better understand diseases of interest and to curate more robust findings for patient care.

**Abstract:**

The development and advancement of aptamer technology has opened a new realm of possibilities for unlocking the biocomplexity available within proteomics. With ultra-high-throughput and multiplexing, alongside remarkable specificity and sensitivity, aptamers could represent a powerful tool in disease-specific research, such as supporting the discovery and validation of clinically relevant biomarkers. One of the fundamental challenges underlying past and current proteomic technology has been the difficulty of translating proteomic datasets into standards of practice. Aptamers provide the capacity to generate single panels that span over 7000 different proteins from a singular sample. However, as a recent technology, they also present unique challenges, as the field of translational aptamer-based proteomics still lacks a standardizing methodology for analyzing these large datasets and the novel considerations that must be made in response to the differentiation amongst current proteomic platforms and aptamers. We address these analytical considerations with respect to surveying initial data, deploying proper statistical methodologies to identify differential protein expressions, and applying datasets to discover multimarker and pathway-level findings. Additionally, we present aptamer datasets within the multi-omics landscape by exploring the intersectionality of aptamer-based proteomics amongst genomics, transcriptomics, and metabolomics, alongside pre-existing proteomic platforms. Understanding the broader applications of aptamer datasets will substantially enhance current efforts to generate translatable findings for the clinic.

## 1. Introduction

Proteomics has continued to establish itself as a field of growing promise both diagnostically and therapeutically. Historically, the two primary methods in proteomic analysis were the enzyme-linked immunosorbent assay (ELISA) and mass spectrometry (MS). ELISA falls short of high-throughput capabilities and faces many challenges in widespread clinical integration due to calibration and quantification, reagent stability/availability/cross-reactivity, biomarker validation, and a lack of validated algorithms for computational analysis [1]. Mass spectrometry emerged over a decade ago as a preferred method for proteomic analysis that could produce large datasets for biological application, predominantly liquid chromatography MS (LC–MS) [2,3]. The ability to analyze post-translational modifications (PTMs) of proteins [4] unleashed a powerful new tool to decode the complexity offered up by evolving MS techniques in extraction, fragmentation, analysis, and database reference. Next-generation high-throughput advances in MS, such as data-independent acquisition MS (DIA-MS), employ permanent digital proteome maps while offering exceptional reproducibility and avoiding the inconsistent precursor ion fragmentation present in large-scale datasets generated via data-dependent MS (DDA-MS) [5]. However, MS suffers from several fundamental drawbacks, including protein inferences in relation to isobaric amino acids, co-elution concerns over PTMs, and flaws in algorithmic database searches [6,7,8]. Established and emerging high-throughput screening approaches [9,10,11,12,13,14,15,16,17,18,19,20,21,22,23] (Table 1) offer a much more comprehensive dynamic range to accommodate the human proteome and produce a higher sensitivity with a much lower detection limit compared to traditional MS platforms [24,25]. Amongst proteomic-based platforms, aptamer-based approaches are growing in number and popularity, generating large-scale datasets (Table 1) [15,26,27]. Aptamers are single strands of oligonucleotides (either ssDNA or ssRNA) that bind with high affinity and selectivity [28]. Selection of aptamers occurs through an in vitro evolution process known as Systematic Evolution of Ligands by Exponential Enrichment (SELEX), in which oligonucleotide libraries undergo multiple automated rounds of positive and negative selection to identify strongly selective aptamers [29,30,31]. These aptamers offer both diagnostic and therapeutic utility. Their highly selective and stable nature offers the potential for robust reproducibility, allowing for the creation of large datasets across multiple experiments. Aptamers have been integrated through various point-of-care diagnostics, synergistic combinations with antibody-based assays, and high-throughput screenings [32,33,34]. However, while sharing the underlying application of aptamers, comparative studies must be undertaken with caution, as even panel-to-panel variance is a commonality. For instance, one of the leading aptamer-based proteomics platforms, SomaLogic, carries several different protein panels ranging from 1300 to over 7000 [9]. Although overlap in protein targets may exist between the various panels, such as the 1.3K [35] and 5K [36] panels, the custom analyte panels may challenge cross-study comparisons. This review addresses the cross-platform intersectionality challenges between aptamers and proteomic “gold standard” platforms such as immunoassay. Aptamer data open a new realm of possibilities in proteomics by overcoming the quantitative volume requirements of LC–MS and exceeding the analyte capacity of ELISA. In addition, aptamer-based technologies are also valuable for application across a wide range of mammalian species, which is helpful for regulatory approval of novel pharmaceuticals which fall under the Food and Drug Administration’s (FDA) Animal Rule. Emerging assays are compatible with a wide range of mammalian species, facilitating simultaneous analysis of samples from human clinical sample repositories and research animal models [9]. Unilateral analyses in omics often fail to capture the complexity of biological biomarkers and struggle with reproducibility. One of the critical challenges in omics analysis is addressing the high dimensionality and cross-referencing of datasets required for independent validation. Omics has made considerable strides in aggregating large-scale databases and repositories for cross-comparisons in genomic and transcriptomic analyses in the last few decades. The establishment of high-quality proteomic data has already begun in LC–MS through the ProCan proteomic knowledgebase, where uniformity in study design, sample preparation, and data analysis is provided [37]. With aptamer-based platforms producing an even more immense amount of data per sample through high-throughput and high-multiplexing analysis, there is a growing need to address the computational analysis of large datasets arising from aptamer technology. Streamlining analysis of aptamer data may expedite the discovery of blood-based biomarkers to surpass the current FDA-approval rate of fewer than two biomarkers per year [38]. Here, we provide a comprehensive review of aptamer-generated data processing and analytical strategies for clinically relevant translation to establish aptamers as a powerful tool in the multi-omics landscape.

## 2. Setting Up Aptamer Studies for Clinical Translation

Aptamer technology provides a means of achieving scalable wide-ranging protein analysis. Samples sent off for aptamer-based panels can consist of various biological matrices ranging from plasma to serum to urine. Requiring roughly around 50 µL of sample volume and possessing more streamlined sample processing workflow techniques [59], aptamer technology can generate highly multiplexed analyses from pre-existing biorepositories that cover more significant concentration gradients than current platforms (Figure 1). Recent studies in the literature back this idea, as archived plasma samples in longitudinal cohorts demonstrate a high protein stability of over 90% at the one-year mark [60]. Accessing existing serum or plasma samples from pre-existing cohorts is another convenient advantage of aptamer-based platforms that enable larger sample sizes. Customizing clinical study designs to cater to the advantages of the aptamer platform will play an essential role in guiding proper statistical analysis for protein effect sizes later. The advantages of utilizing aptamer-based platforms are also associated with unique challenges. While relatively small samples from pre-existing cohorts may be employed, this can result in multifactorial sources of potential sample variability (collection, storage, management of the patients included in the studies) and can result in the utilization of samples that were not necessarily carefully experimentally curated or classified for confounders.

### 2.1. Power, Power Analysis, and Protein Effect Size

Statistical power takes a requisite position at the forefront of study design considerations. Past studies on proteomic-based biomarker discovery reported less than 1% of reported biomarkers being incorporated into commercial assays, with a large number of published studies being underpowered [61]. One of the first National Institutes of Health-led workshops on proteomic biomarker pipelines emphasized the role of outlining basic statistical designs prior to study onset in order to maintain adherence to the FDA guidelines on biomarker translation [62]. A later workshop set forth a quantitative clinical criterion for proteomic biomarkers and a comprehensive statistical design with a requisite recommendation of 0.9 for both discovery and verification stage probability to achieve adequate power [63]. The challenges associated with the broad ranges of abundance in the proteome [64] and ranges of coefficient of variations (CVs) (including biological CVs) were all factored into Skates et al.’s statistical model for sample size. However, aptamer-based platforms may effectively reduce the cohort size requirements of MS-based proteomic platforms given their much lower values of CVs (~5%) [65] compared to technologies, such as multiple reaction monitoring (up to 20%) [66]. Simulations for power analysis factor in CVs, the expected fold change needed, and desired power levels with a recommended four samples per group (CV = 20%, power = 0.8) [67]. Sensitivity and specificity are essential considerations during power analysis, and minimum targets vary according to target clinical use and commonly acceptable risk–benefit analysis [63]. Prospective-based studies striving for aptamer diagnostics require an intricate consideration for sample size and sensitivity and specificity targets in study design [68]. While larger sample sizes are best at increasing statistical power, study-dependent cost and practicality limitations may impede efforts to solicit more samples. Thus, while prospective power analysis is preferred, retrospectively applying power analysis is an option. Previous aptamer-based studies have also leveraged effect sizes (Spearman coefficients and Huber M-Values) to balance small sample sizes [69]. Pilot aptamer studies with limited cohort numbers have depended upon previous reports to estimate experimental models’ area under the receiver operating characteristic curves to achieve requisite minimal power levels of 0.8 [70,71]. However, high-throughput strategies such as aptamer assays process an incredible multiplex of data in the discovery stage such that study design must consider the multiple hypothesis testing problems [72]. Addressing this requires proactive consideration of sample size and statistical analysis methodology to maintain high specificity and sensitivity. While analyses employing platforms with protein analysis capacity in the thousands, e.g., 1.3K, 4K, 7K protein panels [27,61,73], can reveal hundreds of biomarkers for a single disease of interest, the disconnect with translating these biomarkers further down the proteomic pipeline emphasizes the critical need for adequately powered study designs [27,74]. The disconnect is augmented by differential protein expression with respect to the number of samples affected by a significant signal and the extent of the signal itself across samples. These aspects will be discussed in the sections that follow.

### 2.2. Samples and Study Designs: Case–Control and Cohort Studies

Two of the most common choices for designing large-scale proteomic discovery studies are case–control and cohort studies. Case–control [75] enables researchers to access pre-existing biorepositories for clinical samples along with groups that match the disease of interest. However, one of the biggest hurdles facing case–control studies is the inability to compare the disease groups with healthy population controls matched for confounders, such as age, sex, disease-specific factors, and comorbidities. Matched (individual or frequency) case–control studies represent one route of generating such controls but can introduce new biases. The proteome is particularly vulnerable to conditions such as organ trauma, autoimmune disease and age, in particular, as pertaining to the analysis of samples originating from adults or a pediatric cohort as well as ethnicity and sex differences. While steps are being taken to employ reference normalization that accounts for some conditions and demographic aspects using robust reference normalization, this is an area of active evolution in aptamer datasets. Comparisons between matched and unmatched designs report minimal outcome distinctions aside from a slight decrease in statistical power for matched samples [76]. Care must be taken to address and reduce the selection bias that commonly occurs in single-institution case–control studies. For matched studies, conditional logistical regression is the mainstay analysis tool, but unconditional logistical regression has also been simulated to mirror these results [77] closely. A less common primary approach involves cohort studies that struggle with extended time frames, costs, and incomplete datasets due to patient dropouts. However, cohort studies may play a vital role later in aptamer-guided biomarker discovery due to their ability to access pre-existing longitudinal datasets for independent cohort cross-referencing [78]. Assay datasets can be optimized for analysis, ensuring normalization for inter- and intraplate variance with internal controls. Hybridization normalization, well-to-well standardization, signal calibration, and standard operating procedures support reproducibility. Thus, differential expression can then be determined based upon comparisons of relative fluorescence unit (RFU) ratios like multiplex bead-based assay data. The following section highlights fundamental tools in transforming these large datasets into workable biological knowledge and translatable hypotheses.

## 3. Statistical Strategies for Analyzing Differential Expression

Aptamer-based platforms hold great promise within the field of shotgun proteomics and carry similar considerations as MS-based platforms in categorizing differential expression with the added advantage of high-throughput analysis of multiple samples simultaneously. Multimarker-based approaches usually tend to overfit, leading to conceptual errors. Statistical analyses should include an adjustment for multiple testing. Multiple biomarkers can be combined in a classifier that outperforms a single biomarker, and validation in an independent sample is imperative, which gives more confidence in the results. Failures of small studies to detect biomarkers often result from variability that interferes with determining effect sizes. Therefore, an increase in both samples and biomarkers contributes to developing multimarker classifiers with enhanced accuracy. Prospective-based studies striving for aptamer diagnostics require an intricate consideration for sample size, sensitivity, and specificity of the targets in study design [33]. A protein may exist in multiple forms within a cell or cell type. These protein isoforms originate from transcriptional, post-transcriptional, translational, post-translational, regulatory, and degrading and preserving processes that affect protein structure, localization, function, and turnover. The field has thus evolved to include a variety of methods for the separation of complex protein samples followed by identification using proteomics technology. It is inherently a systems science that considers protein abundances in a cell and the interplay of proteins, protein complexes, signaling pathways, and networks. In order to address the relevant challenges, the analytical tools can be categorized into four types: (1) quality control, (2) fundamental statistical analysis, (3) machine learning (ML) approaches, and (4) assignment of functional and biological information to describe and understand protein interaction networks [79].

### 3.1. Quality Control and Basic Statistics

Quality control (Table 2) is employed to observe the data variability, compare means between groups, and look for any anomaly that could cause a problem in the analysis. Quality control can identify significant areas of concern and flag samples as well as more specifically diagnose sources of potentially aberrant data signals; however, detailed capture and annotation is required at all levels from sample collection to analysis to ensure that robust data are obtained and analyzed for confounding features. Basic statistics are a critical first pass to identify the “low-hanging fruit” in the dataset. Methods such as the Student’s *t*-test and its nonparametric equivalent, the Wilcoxon test, univariate or analysis of variance (ANOVA), or the nonparametric Kruskal–Wallis test are applied to identify the significant proteins. Due to inherent variability, statistics alone are often insufficient to discover the most biologically relevant information in a proteomic dataset but they are an essential first step in every analysis. Statistically significant results are helpful as seed data or bait in machine learning approaches.

### 3.2. Machine Learning Approaches

Classification by ML (Table 2) complements traditional statistics as it allows for consideration of many variables at once and removes much of the technical bias. Dataset complexity is reduced, as correlations and trends are identified that may not be visible or may be undetectable using traditional statistics, e.g., clustering using iterative subsampling. Given unbiased data inputs, ML classification has the potential to be unbiased by revealing patterns within data that may or may not relate to the original hypothesis. The researcher is then able to examine the clustering or classification results for new biological features that were not initially predicted. Thus, ML together with network tools enable hypothesis generation, as they uncover the real biology of the system in question. Swan et al. [94] discussed the benefit of ML methods for application to proteomic data and show that machine learning methods give an overall view of data and offer a large potential for identifying relevant information among data. While ML approaches have been more extensively employed to advance aptamer discovery [95], the use of ML, deep learning (DL), and artificial intelligence (AI) applied to large-scale data originating from aptamer studies to arrive at clinically meaningful and relevant conclusions is still in its beginnings. Broadly, artificial intelligence methods can involve classic ML, using techniques such as support vector machines (SVM) and random forests (RF). Alternatively, DL with convolutional neural networks (CNNs) and hybrid techniques of both classical ML and DL may be employed [96]. Generally, AI methods may be supervised (where the model is told the outcome of interest) or unsupervised, where the model does not know the outcome [97]. Supervised approaches require the data to be divided into categories and training and testing sets, with the model trained on a portion of the data and tested on the remainder. As employed in DL, neural network models do not typically require annotation, but the process by which the model arrives at the results may be difficult to elucidate, i.e., a “black box.” ML is currently evolving to analyze aptamer technology-generated data in conjunction with other data types, including clinical, imaging, pathology, and other omics data. A plethora of proteomic alterations are identified using aptamer technology with multiple and many as yet poorly understood signals; hence, aptamer data can lend itself to AI approaches to connect results to clinical meaning. Discussion is ongoing on the optimal means of analysis. Although specific examples of AI as applied specifically to aptamer technology-derived data are scant, the literature originating in MS and RNA sequencing offers more in-depth explorations [97], and in non-aptamer technology, proteomic data [98] parallels exist. Specific known protein alterations may be identified, analyzed, and then extrapolated to other related proteins using known or evolving signaling pathways. Alternatively, emphasis can be placed on filtering out pertinent signals using artificial intelligence approaches and then connecting these to known and unknown proteins and clinical data. Currently, there is a lack of ground truth in aptamer data, limiting the ability to validate findings and train DL methods that are traditionally data-hungry. There is also a lack of standard datasets to provide a reliable comparison for abnormal samples. Nonetheless, creating clinical connections using aptamer data is undergoing active progress across medical disciplines [35,99,100,101]. The goals of aptamer-based data intersecting with AI currently focus largely on diagnosis [35,99,100] evolving into response assessment [101], with few publications exploring ML to examine management or prognosis. In a diagnostic example, using urine samples, Dong et al. employed the SOMAscan platform to identify culture-positive urine samples in the urine of 16 children with urinary tract infections. ML with SVM based feature selection was performed in this study to determine the combination of urine biomarkers that optimized diagnostic accuracy [99]. The authors found that eight candidate urine protein biomarkers met filtering criteria resulting in area under the receiver operating characteristic curves (AUCs) ranging from 0.91 to 0.95, with the best prediction achieved by the SVMs with a radial basis function kernel [99]. In the context of arthritis, the serum proteome for patients with psoriatic arthritis and patients with rheumatoid arthritis was analyzed using nano-liquid chromatography–mass spectrometry (nano-LC–MS–MS), SOMAscan, and Luminex, and multivariate ML was employed on the data from all three platforms to separate patients with early-onset inflammatory arthritis to differentiate psoriatic and rheumatoid arthritis [100]. In the context of sleep apnea, Ambati et al. employed the Obstructive Apnea Hypopnea Index (OAHI), the Central Apnea Index (CAI), the 2% Oxygen Desaturation Index, and mean and minimum oxygen saturation indices during sleep to train a machine learning classifier using a SOMAscan 1.3K assay and achieved 76% validation accuracy [35]. Hewitson et al. also used SOMAscan and machine learning to identify nine proteins that were significantly different in autism spectrum disorder vs. typically developing boys, although the authors acknowledged that further verification with independent test sets is warranted [102]. In an example of aptamer data and AI applied towards response assessment, O’Neil et al. studied clinical remission in rheumatoid arthritis using 130 patient serum samples on a 1.3K SOMAscan platform. They employed unsupervised hierarchical clustering and supervised classification to identify proteomic-driven clusters for model biomarkers associated with future disease flare after 12 months of follow-up and medication withdrawal. Network analysis was employed to define pathways that were enriched in proteomic datasets. The authors found that clustering did not predict future risk of flare, while the XGboost machine learning algorithm classified patients who relapsed with an AUC of 0.80 using only baseline serum proteomics [101]. Machine learning and AI methods are actively evolving, and their application to aptamer-based data is currently limited but growing as the technology is more widely applied. Most methods encountered currently involve classical ML and significant annotation. In the absence of robust controls and significant annotation, ML has been more often applied to aptamer technology-derived data in the context of diagnosis, as discussed above. Therapy-related questions require robust controls, which are difficult to obtain as patients undergo heterogenous management over time in most settings, which is expected to modify the proteome.

### 3.3. Pathway Analysis

Pathway analysis (Table 2) following statistical analysis, classification, and clustering can help organize a long list of proteins onto a short list of pathway knowledge maps, easing interpretation of the molecular interplay. The machine learning and clustering tools of omics data can be categorized into a supervised and unsupervised classification for seven popular types of machine learning: principal component analysis (PCA), independent component analysis (ICA), K-means, hierarchical clustering, partial least squares (PLS), random forests (RF), and SVMs. These methods are also summarized and compared in Table 2, which provides an overview of different classifications and clustering tools and how to select a method most likely to be effective for a specific dataset. The intersection of high-throughput, high-multiplex proteomic datasets, existing omics databases, and clinical features results in a rich systems biology analysis to better understand biological pathways and functional gene networks. Seated within systems biology is pathway analysis. Pathway analysis facilitates future hypothesis generation from high-throughput microarray data, localizing gene networks, and framing protein differential expression into meaningful nodes and modules. Understanding how aptamer-identified proteins operate interconnectedly can support biomarker identification and further the identification of aberrant biological pathways in disease [103]. The underlying premise of pathway analysis involves preparing and standardizing protein differential expression from aptamer data (typically evaluated via fold-changes) [104], performing statistical analysis of relevant proteins, and applying a pathway database. There are three generations of pathway analysis: over-representation analysis (ORA), functional class scoring (FCS), and pathway topology (PT) [105]. ORA was the first and most simplistic form of pathway analysis. After initial statistical analysis, aptamer datasets reveal a list of statistically significant proteins that are either over- or under-expressed. Accompanying these proteins is a list of genes inputted into ORA to measure the most over-expressed genes via hypergeometric analysis, producing over- or under-represented pathways based upon a previously selected FDR. ORA was one of the first widely used pathway analysis strategies and has even been utilized in aptamer-based studies of the SARS-CoV-2 virus [46]. However, this inherent independent-based assumption of genes has generated high false-positive rates due to the correlations between genes that are ignored in ORA [106]. Furthermore, the arbitrary cutoff threshold can significantly impact the conclusions drawn [107]. Thus, while ORA presents a simple, cost-effective option for analyzing aptamer datasets, the fundamental limitations of single-set gene analysis restrict the technique’s robustness. Functional class scoring, a second-generation approach, utilizes a three-step process of computing gene-level statistics, compiling them into pathway-level statistics, and assessing for statistical significance [105]. FCS overcomes the arbitrary thresholds utilized in ORA and further takes into consideration subtler changes and impacts of coordinated networks. However, recent studies have faulted FCS for its lack of specificity [108]. Gene set enrichment analysis (GSEA) is one of the most popular gene set analysis techniques under FCS and it has seen widespread use in aptamer studies, including discoveries in Duchene muscular dystrophy [109], myocardial infarction [47], and myeloid leukemia [110]. GSEA leveraged past research and was one of the first methodologies to focus analysis on deriving an understanding of gene sets [89]. GSEA has also given rise to parametric analysis of gene set enrichment (PAGE), which potentially offers a more sensitive analysis while avoiding the rigorous computational effort required in GSEA [111]. A third method known as generally applicable gene set enrichment (GAGE) has also emerged to tackle datasets of different sample sizes, which may be more applicable towards cross-validating aptamer studies of different experimental designs or methodologies [112]. Cross-comparisons of analysis techniques suggested that GAGE had the highest reproducibility and predicted the most relevant gene sets [113]. Finally, the third-generation approach known as pathway topology (PT) mimics the three-step process of FCS but applies pathway topology in computing gene-level statistics [105]. PT can consider the interactions between genes and avoid the independency assumptions about genes in FCS. A recent characterization of all pathway analysis methods gives PT-based approaches a slight edge over non-PT-based approaches in relation to real-world data [114]. Since the onset of pathway analysis, dedicated pathway analysis databases have emerged that support the distinct subcategorizations of pathway analysis described above. Popular databases are highlighted in Table 2, including Gene Set Enrichment Analysis (GSEA) [89], Ingenuity Pathway Analysis (IPA), the Database for Annotation, Visualization, and Integrated Discovery (DAVID) [90], Cytoscape [91], the Kyoto Encyclopedia of Genes and Genomes (KEGG) [92], and Human Annotated and Predicted Protein Interactions (HAPPI) [93]. A full comparison of current databases was recently conducted by Chowdhury and Sarkar [115]. One of the most comprehensive platforms currently available is QIAGEN’s IPA program that features support for protein interactions, metabolic data, gene regulation, and sequencing [116]. IPA has facilitated extensive pathway analysis in aptamer datasets in atrial fibrillation [117], surgical procedures [48], and aging [118]. While pathway analysis is a fundamental step in deriving meaning from the differentially expressed proteins derived from aptamer technology, it is incumbent on researchers to understand this area’s current challenges and limitations. First, pathway analysis and databases rely on a body of published literature that continuously evolves and adapts. Second, meta-analyses of the published literature suggest that poor concordances exist based upon the type of pathway analysis selected and the database utilized [119,120]. As a result, care must be taken in selecting the best suitable methodology and database based upon the study design. The ability to create and validate pathway-specific reference panels for different clinical contexts while employing aptamer-based data will improve the ability to design studies as proteomic “controls” become more prevalent.

## 4. Multi-Omics Approaches and Verification

### 4.1. Proteomic Quantitative Trait Loci (pQTL)

While pathway analysis provides an excellent canonical interpretation of protein groups, the next step in verifying aptamer microarrays’ proteomic data is to establish genetic anchorage for identified proteins. One of the emerging methods to accomplish this is through proteomic quantitative trait loci (pQTL), which evaluates the variance in proteins attributable to specific loci. pQTL has recently supplanted expression quantitative trait loci (eQTL) due to the latter’s dependency on the poor relationship between mRNA expression and protein levels, as well as a reported disconnect of roughly 50% between pQTL and eQTL [121,122]. The incredible scalability of aptamer microarrays enables studies to connect large-scale proteomics data with pQTLs. Recent applications of aptamer microarrays have already been made in studying the proteome of patients infected with the recent SARS-CoV-2 virus [123]. In turn, pQTLs can substantiate cross-platform findings between aptamer platforms and others which can impact future translational capabilities of studies. Pietzner et al. reported several factors leading to aptamer-specific pQTL findings, such as lower observational correlations, lower binding affinities of aptamers, and extreme datapoints that merit consideration in the verification process [33]. pQTL takes on a fundamental role in verifying proteomic relationships hypothesized from discovery-stage analysis. After multi-fold reductions take place to isolate a small number of differentially expressed proteins, these proteins must be properly validated before being passed on as potentially viable biomarkers. One of the primary benefits of applying pQTL analysis is to establish an integrative understanding of causal networks and pathways by combining large-scale databases of both aptamers and genome-wide association studies data (GWAS) [124]. Ferkingstad et al.’s 4.9K SOMAscan protein assay identified nearly 18,000 associations of sequence variants and pQTLs and applied a multi-omics method of proteomics, transcriptomics, and genomics towards a large-scale aptamer database [125]. Furthermore, the authors emphasized associations of variants with high LD or *cis* (near the gene of interest) pQTLs in pursuit of developing drug targets [125]. A separate large-scale SOMAscan-based study (INTERVAL) also previously established the role of Mendelian randomization analysis for applying aptamer-based proteomic datasets towards discovering causal protein biomarkers [126]. Studies have also applied aptamer datasets and pQTLs with GWAS, mRNA, and eQTL datasets to link proteomic variability with genetic components in a cohort of irritable bowel syndrome patients [127]. Population samples have also been utilized in combining aptamer datasets with pQTLs to demonstrate utility and verification [128]. Aptamer platforms have also supported large-scale curated datasets which have been employed in “virtual proteomics” for clinical prediction and biomarker discovery [129].

### 4.2. GWAS and PWAS

Protein-wide association studies (PWAS) represent a newly emerging analysis tool in the multi-omics landscape that attempts to establish an understanding of combining GWAS with protein functions and phenotypes. Simulation testing suggests that PWAS is better for causal relationship analysis, has a reduced computational burden, is complementary to SKAT (GWAS), and can find PWAS-exclusive genes [130]. Applied to large-scale databases for Alzheimer’s disease, PWAS uncovered additional AD genes of interest not found through traditional GWAS [131]. Summary data with GWAS can be performed using summary data-based Mendelian randomization and heterogeneity in dependent instruments (HEIDI) [132]. This growing integration between aptamer datasets, genomics, transcriptomics, and proteomics will continue to evolve as aptamer technology matures and scales in magnitude. Significant effort is directed at identifying translatable protein biomarkers while establishing a robust genetic anchorage for differentially expressed proteins, with the understanding that doing so is vital for clinical impact with eventual prospects for FDA approval [133].

## 5. Translational Challenges of Aptamer Proteomics

While the technical capabilities of aptamers with their flexibility, high multiplexing, and strong affinities open a wealth of opportunities both diagnostically [134] and therapeutically, only a minimal number of studies have seen success clinically. This review centers around examining data arising from aptamer-based assays used in biomarker and clinical diagnostics. Although SOMAscan remains one of the most popular platforms for high-throughput proteomic analysis of biological matrices, other proteomic aptamer platforms continue to be developed, such as ProtSeq [10], protein precipitation assays [135], and disease-specific panels [136,137]. However, aptamer-based proteomics has yet to be promoted to mainstream prominence as in the case of mass spectrometry platforms. We predict that the growing capabilities of aptamers will lead to a continual acceleration of aptamer platforms that have only recently been developed. As evidenced by mass spectrometry proteomics, aptamer-based proteomics will likely face similar challenges. Here, we cover challenges in proteomic biomarker translation and cross-platform consistency.

### 5.1. FDA Approval and Clinical Translation

Aptamer-based platforms embody the beginning stages of the traditional biomarker discovery pipeline (discovery, verification, and validation), sharing a position with mass spectrometry [138] (Figure 2). One of the common challenges in proteomic biomarker discovery is the poor translation into the clinic, with an extensive track record of biomarkers fizzling out during FDA approval [139]. While the high multiplexing capability of aptamers broadens the survey of the human proteome, the sheer number of proteins and their varying abundances make detecting disease-specific variations a challenging feat. As aptamer assays continue to broaden over time, a more encompassing picture may be drawn at the cost of higher false discovery rates. With the arduous task of achieving FDA approval [140], false discovery rates of inaccurate biomarkers come at a high cost. While adequately designed statistical analysis and sufficiently powered studies may address this, single protein-based biomarkers will merit a considerable effort to achieve approval. Multi-marker panels serve as a more robust substitute [141,142]. Two significant advances in proteomic biomarker discovery in the last decade have been the FDA approval of the OVA1 [143] and CKD273 [144] panels. The CKD273 panel was based upon capillary electrophoresis coupled to mass spectrometry (CE–MS) technology, which supports the throughput of thousands of peptides that strongly supported earlier stages of research [145]. Building large, comparative datasets that feature large sample sizes can greatly support discovery-stage proteomics. The high precision, throughput, and multiplex aspects of aptamers facilitate this goal. It is paramount for studies to consider analytical factors for effective translational hypothesis generation to follow. Current ambitions for blood-based diagnostics and liquid biopsies in cancer [146] will continue to fuel a strong interest in aptamer-identified biomarkers. Clinical translation, however, will depend on the ability to transfer findings between the clinic and bench research, which will require that subsets of large-scale aptamer-based data panels be transferrable between species to allow for findings to be replicated in laboratory animals and human samples to advance outcomes predicated on alteration in management.

### 5.2. Cross-Platform Consistency

While aptamers can generate a large dataset based upon only a few clinical samples, these data may not substitute for a multi-omics approach. Single-omics analysis is an inherent limitation of aptamer assays, as protein expression levels may not be an accurate indicator with transcript levels and other omics data [147]. Previous studies have suggested a poor correlation between aptamer- and antibody-based platforms, which may limit the biomarker discovery potential of aptamer-based technology [148]. Cross-platform comparisons across the same cohorts have found wide-ranging correlations [149]. However, poor concordance may not suggest inefficacy. While inter- and intra-platform variations exist, a recent study in cardiac patients reported that all statistically significant proteins identified via aptamers were similarly identified in immunoassays [150]. Part of this variation may be attributable to aptamer technology’s intrinsic ability to measure more extensive dynamic ranges not accounted for in techniques such as mass spectrometry [151]. Additionally, reproducibility studies with aptamers and ELISA have suggested a relatively high concordance rate between the two technologies [49]. Reports in the literature suggest that, despite some differences occurring between platforms, aptamers and immunoassays essentially point towards similar biomarker–disease associations, with higher biomarker concentrations leading to more robust findings [150]. While these studies are encouraging, cross-platform concordance remains a limitation of aptamer-based data. Thus, while care should be taken in drawing cross-platform comparisons and interpreting meta-analyses, aptamer technology should not be interpreted in isolation but rather seen as a complementary tool to current proteomic technology.

### 5.3. Intra-Platform Consistency

Intra-platform consistency should also be considered, for instance, as earlier versions of the SOMAscan assay specific for approximately 1300 or 4100 [73] protein targets may not be compared directly with the latest assay version specific for approximately 7000 protein targets. Though expression trends may be comparable, the raw numerical values for each specific aptamer may change across assay versions due to technical variables, and each specific assay does not necessarily include the same cohort of aptamers, which has implications for efforts to leverage for comparison the findings from previous clinical studies which have utilized different versions of the assay. This phenomenon is also not limited to the SOMAscan technology, as other multiplexed proteomics platforms of all denominations (1.3K, 4K, 7K) [48,51,61,73] face the same challenges of compatibility across datasets as assay specifications change over time. Of note, a major Human Protein Atlas study tested over 9000 internally generated antibodies using immunoassays, finding that half failed to match literature results and only 7% demonstrated strong concordance with the literature [152]; similar challenges are anticipated in the context of aptamer-based data.

## 6. Conclusions

Aptamer technology generates extremely large datasets that are growing in number and popularity. The ability to harness a small amount of biospecimen and the potential for clinical applicability and aggregation with existing and evolving pathway analysis options make large scale aptamer-based datasets particularly attractive to researchers in all areas of medicine. Ongoing emphasis is being placed on study design and statistical considerations for analysis. However, unravelling the complexity of the human proteome will continue to pose a substantial challenge to translating large datasets into clinical utility. Practical guidance for planning aptamer-based studies includes:
(1)Study designThe study design for aptamer-derived data is understandably contingent on the disease and clinical context. Maximal benefit is elicited if the study benefits from maximal data capture and annotation to ensure that potential confounders in the proteomic data signals can be addressed down the line. Controls are crucial for meaningful comparison. Controls may not necessarily represent a normal population but rather a population whose proteome in comparison to that of the intervention will allow the researcher to derive conclusions that will address the hypothesis being tested. Cohort studies may be employed with a “before” and “after” sample being obtained from the same patient, thus using the patient as their own control. Provided robust large-scale data are available and, following a thorough review, are sufficiently comparable to the study population, this is a reasonable option, with the study design in this context benefitting greatly from collaboration amongst researchers.(2)Statistical analysisStatistical analysis for aptamer-derived data is actively evolving. Traditional approaches described here may be employed. ML approaches may be used with semi-supervised approaches likely to be the most successful and contingent on annotation of the data.(3)An established team that combines researchers, statisticians, and clinicians will need to maintain a close relationship in the planned acquisition, curation, annotation and analysis of data to allow for meaningful translation into the clinic and advancement in relation to patient outcomes.

## Figures and Tables

**Figure 1 cancers-14-02227-f001:**
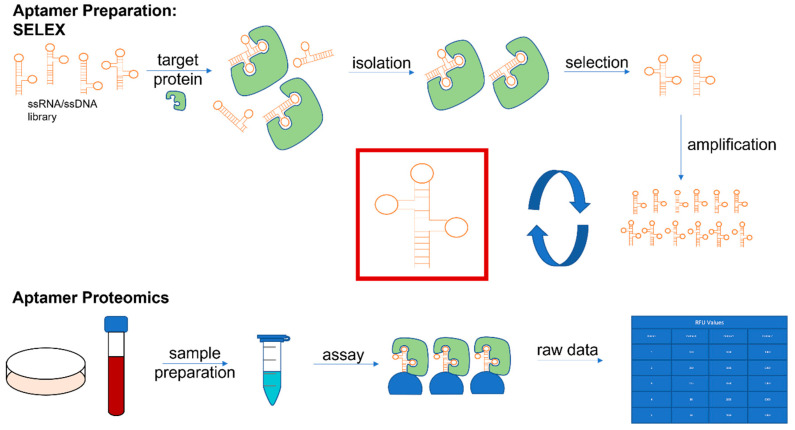
Generating proteomic datasets using aptamers.

**Figure 2 cancers-14-02227-f002:**
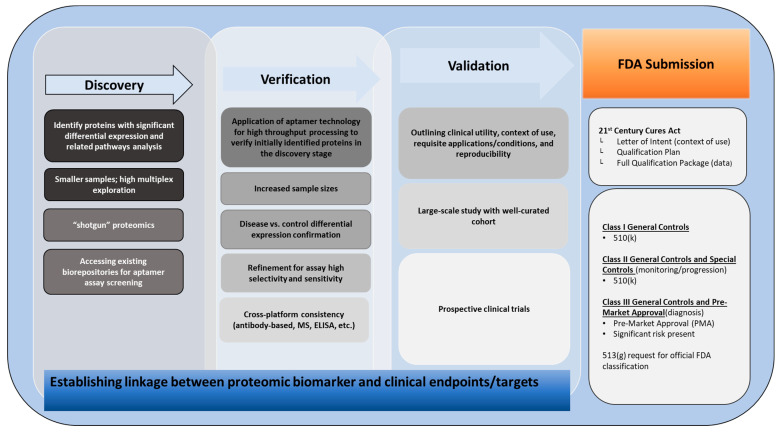
Process map advancing large-scale proteomic datasets from discovery to verification to validation towards identifying clinically meaningful biomarkers for FDA submission.

**Table 1 cancers-14-02227-t001:** Overview of Common Proteomic Platforms.

Analytical Technique	Category	Protein Sample Literature Values └	Accepted Biospecimen Types	Reported Dynamic Range ‡	CV └└	Protein Capacity Multiplex)
**Proximity Extension Assay (Olink)** [11]	Antibody	1 µL	Plasma, tissue/cell, synovial fluid, CSF, plaque extract, and saliva	LLOQ = 0.25 pg/mL	7.8% (intra) and 10.6% (inter)	***
**Reverse Phase Protein Arrays** [12,39,40,41,42]	Antibody	5 µg (1.0 to 1.5 mg/mL protein)	Tissue/cell, plasma, serum, biopsies, body fluids	LOD = 0.55 fg/mL	<15%	***
**Bio-Plex** [13,43]	Antibody (bead)	12.5 µL (serum/plasma) 50 µL (cell culture)	Plasma, serum, tissue/cell	LOD = 0.6–6.4 pg/mL	2–15%	****
**Simoa** [14,44,45]	Antibody (bead)	25 µL	Plasma, serum, urine, tissue/cell, CSF, saliva	LOD = 0.005 pg/mL	<10%	***
**Aptamer Group (Optmer)** [15]	Aptamer	38 µL	Plasma (diagnostics and therapeutics), urine, tissue/cell, liquid matrices	LOD = 55 ng/mL	<5%	*
**Base Pair Technologies** [16,26]	Aptamer	5–100 µL	Plasma, serum, tissue/cell	LOD = 1 pg/mL		**
**SOMAscan** [9,27,46,47,48,49]	Aptamer	55–100 µL	Plasma, serum, CSF, urine, cell/tissue, synovial fluid, exosomes	LOD = 1.6 pg/mL	4.6%	******
**Electrochemiluminescence Immunoassay (Meso Scale and Lumit)** [17,50]	ECLIA	50 µL	Plasma, serum, tissue/cell, CSF, urine, blood spots, tears, synovial fluid, tissue extracts	LOD = fg/mL	5–10%	**
**Multiplex ELISA** [1,18,43,51,52]	ELISA	25–50 µL	Plasma, serum, tissue/cell, urine, saliva, CSF	LOD = 0.61 to 18.90 pg/mL	9.5–28.5% (inter/intra)	**
**Singleplex ELISA** [53]	ELISA	100 µL	Plasma, serum, tissue/cell, urine, saliva, CSF	LOD = pg/mL	1.6–6.4% (intra) and 3.8–7.1% (inter)	*
**2D-PAGE** [19]	Gel electrophoresis	~100 µg (15–50 µL)	Plasma, serum, tissue/cell, urine	LOD = 10 ng to 100 ng	<20%	******
**DDA-MS** [20]	MS	10 µL	Plasma, serum, tissue/cell	LOD = 157 ng/mL	5.7%	*****
**SWATH-MS** [21,54,55]	MS (DIA)	5–10 µg	Plasma, serum tissue/cell, platelets, monocytes/neutrophils	LOD = 1 fmol	13.7%	*****
**iTRAQ** [22,56]	MS (labeling in LC–MS–MS)	12 µg	Plasma, serum, tissue/cells, saliva	LOD = 1 fmol (50 µg/mL)	<0.53%	*****
**SRM/MRM** [23,57,58]	MS (LC–MS–MS)	15 µL	Plasma, tissue/cell, dried blood spots	LOD = µg/mL (no enrichment)	6.1% (intra) and 11% (inter)	**

**└** Actual requirements may vary depending on transit conditions, company selected, and number of panels desired. **└└** Values reflect reported literature values; technical specifications vary based upon instrument and sample conditions. Groupings: * = 1–10; ** = 10–100; *** = 100–500; **** = 500–1000; ***** = 1000–5000; ****** = 10,000+. ‡ Quantification subject to multifactorial variability secondary to data origin, methodology, assay type, and analytes tested.

**Table 2 cancers-14-02227-t002:** Statistical methods for aptamer analysis.

Quality Control		
Student’s *t*-test or nonparametric Wilcoxon	Mean difference between two groups	[80]
ANOVA or nonparametric Kruskal–Wallis	Variation between two or more groups	[81]
Visualization methods	Histogram, density plots, box and bar graphs	
**Classification**		
Principle Component Analysis (PCA)	Dimension reduction, separates groups based upon commonality	[82]
Independent Component Analysis (ICA)	Dimension reduction, separates groups based upon correlation	[83]
Partial Least Squares (PLS)	Discriminant analysis that separates groups by maximum covariation ranks the important features	[84]
Random Forest (RF)	Separates groups by similarity, ranks important features	[85]
Support-Vector Machine (SVM)	Classifies the sample by kernel function	[86]
**Clustering**		
K-means	Clustering of features or samples into user-specified numbers of clusters	[87]
Hierarchical	Unsupervised classification of features, samples, or any endpoint by dendrogram	[88]
**Pathways**		
Gene Set Enrichment Analysis (GSEA)	Pathway analysis and functional annotation	[89]
Ingenuity Pathway Analysis (IPA)	Pathway and functional annotation from curated databases	
Database for Annotation, Visualization, and Integrated Discovery (DAVID)	Pathway and functional annotation using Gene Ontology (GO)	[90]
Cytoscape	Network analysis visualization	[91]
Kyoto Encyclopedia of Genes and Genomes (KEGG)	Pathway analysis	[92]
Human Annotated and Predicted Protein Interactions (HAPPI)	Protein interactions	[93]

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
