# Peer review of "Analytical Considerations of Large-Scale Aptamer-Based Datasets for Translational Applications"

_cancers, 2022, doi:10.3390/cancers14092227_

Round 1

Reviewer 1 Report

As a reviewer, I read the manuscript entitled "Analytical Considerations of Large-Scale Aptamer-Based Datasets for Translational Applications" with great interest, and at the same time I appreciated its relevance and accessible form of presentation. I gained certain new knowledge, found good structuring of the material and recognized the sufficient competence of the authors in the issue under consideration.
The manuscript considers comprehensive issues relevant to the discovery, validation and implementation of new biomarkers using aptamers. Various proteomic platforms and aspects of their translation into the clinic are considered. Statistical strategies for interpreting proteomic data are presented from an interesting perspective. The review appears to be comprehensive and self-contained, ready for publication in its current form.

Perhaps it lacks some comparative details of the described multiplex aptamer systems, which may be present in the cited sources. There are no details of the development and production of such powerful multiplex aptamer systems, as their problems and shortcomings are not explicitly discussed in direct comparison with the mass spectrometric proteomic approach. There is also no direct comparison of existing multiplex aptamer systems. The ability to quantify the presence of a biomarker using certain aptamer panels (as well as the general question of quantification with aptamers) is not touched upon in detail. But, perhaps, these questions are beyond the scope of the tasks set by the authors for the presented review, which already looks quite sufficient and interesting for the audience of the journal in the presented form.

Author Response

We agree with the reviewer and have added content to address the points raised here including by adding content to highlight the challenges in setting up studies which are indeed significant and evolving and guidelines to aid in planning aptamer-based studies. Our references do indeed address the aspects raised regarding comparison of platforms and this remains a limitation which we have highlighted in more depth. 

Reviewer 2 Report

The review relates to a very interesting and up-to-date topic in the aptamer field, aptamer-based proteomic technologies. The manuscript summarizes recent publications in the field, analyzes current trends and points to the problems of aptamer datasets applications in proteomics which still remain to be solved. The manuscript is well-organized and well written and can be accepted in its current form. The only very minor comment concerns the SELEX abbreviation (page 2, line 57): Systematic Evolution, not Sequential (see Tuerk and Gold, 1990).

Author Response

We agree and have corrected this. 

Reviewer 3 Report

In this manuscript, Jiang et al. reviewed the application of aptamer technology in proteomics and biomarker discovery. They highlighted the advantages of aptamer-based mass spectrometry, including high throughput, specificity, sensitivity, and multiplex. In addition, they discussed some challenges, such as a lack of a standardizing methodology for analyzing the large datasets using the aptamer-based approach. Additionally, they addressed some analytical considerations of surveying the initial data, deploying proper statistical methodologies to identify differential protein expressions, and applying datasets to discover multi-marker and pathway level findings. Finally, they discussed aptamer datasets within the multi-omics landscape by exploring their intersectionality amongst genomics, transcriptomics, and metabolomics.

Overall, it is a well-written, comprehensive review. However, there are still some limitations. It is not clear what the unique challenges of aptamer-based approaches are and how to address them. The methodologies for power analysis, samples and study designs, and statistical strategies discussed in this manuscript are currently used in the proteomics field. What are special considerations when they are used in aptamer-based studies? In other words, what proteomics scientists should have done differently from the current proteomics platform when they plan to adapt aptamer-based approaches. This review did not provide clear and practical guidance on how to plan an aptamer-based study and carry out the downstream analysis. Furthermore, the review did not discuss the limitations of the aptamer-based approach and how to address them.

Author Response

  • It is not clear what the unique challenges of aptamer-based approaches are and how to address them.

We agree with the reviewer and have added content to highlight the challenges in  setting up studies, validating data and producing transferrable results all of which which are indeed significant and evolving.

  • The methodologies for power analysis, samples and study designs, and statistical strategies discussed in this manuscript are currently used in the proteomics field. What are special considerations when they are used in aptamer-based studies?

We have added content to highlight considerations particularly affecting aptamer studies, including as related to sample acquisition, data collection and validation.

  • In other words, what proteomics scientists should have done differently from the current proteomics platform when they plan to adapt aptamer-based approaches. This review did not provide clear and practical guidance on how to plan an aptamer-based study and carry out the downstream analysis.

We agree with the reviewer and have added content to aid in planning aptamer-based studies to offer additional guidance.

  • Furthermore, the review did not discuss the limitations of the aptamer-based approach and how to address them.

We have added content to highlight limitations affecting aptamer studies, which are actively evolving.